# Mechanochemical Transformations of Polysaccharides: A Systematic Review

**DOI:** 10.3390/ijms231810458

**Published:** 2022-09-09

**Authors:** Tatiana A. Akopova, Tatiana N. Popyrina, Tatiana S. Demina

**Affiliations:** Enikolopov Institute of Synthetic Polymeric Materials, Russian Academy of Sciences, 70 Profsouznaya Str., 117393 Moscow, Russia

**Keywords:** solid-state organic reactions, mechanical activation, polysaccharides, derivatives, graft copolymers, composites, biocompatible materials, environment friendly technologies

## Abstract

Taking into consideration the items of the Preferred Reporting Items for Systematic Reviews and Meta-Analyses (PRISMA), this study reviews application of mechanochemical approaches to the modification of polysaccharides. The ability to avoid toxic solvents, initiators, or catalysts during processes is an important characteristic of the considered approach and is in line with current trends in the world. The mechanisms of chemical transformations in solid reactive systems during mechanical activation, the structure and physicochemical properties of the obtained products, their ability to dissolve and swell in different media, to form films and fibers, to self-organize in solution and stabilize nanodispersed inorganic particles and biologically active substances are considered using a number of polysaccharides and their derivatives as examples.

## 1. Introduction

### 1.1. Rationale

The mechanochemical approach is based on the treatment of a solid mixture of reagents under the combined action of pressure and shear strains in apparatuses of various types, which leads to disordering of their supramolecular structure and chemical reactions induced by mechanical energy absorption. The largest number of works devoted to the study of changes in the physicochemical properties of solids under mechanical action has been performed for inorganic reactive systems using ball, planetary, vibration or other mills—pulsed action devices [1,2]. The mechanisms of activation of solid organic and inorganic substances to chemical transformations under mechanical action have common patterns [3]. However, the practice of mechanosynthesis, which is widely used on an industrial scale for many inorganic substances and mixtures of metals, encounters severe limitations in organic synthesis due to the low thermal stability of organic compounds and the duration of relaxation processes [4,5]. In the last decade, there has been a significant increase in interest in the processes occurring in polymers under mechanical action [4,6,7]. And although the authors note the possibility of the formation of new covalent bonds after breaking during mechanical destruction of “weak” bonds in the structure of macromolecules and the existence of so-called “mechanophores”—groups responsible for selective scission and capable of isomerization under mechanical loading, nevertheless, full-scale work on carrying out polymer-analogous transformations of polymers during their interaction with solid reagents of various nature are still at the beginning phase. At the same time, progress in the field of methods for studying the structure of polymers subjected to mechanical activation allows us to hope for the development of ideas about the possibilities of mechanochemistry for purposeful modification of their structure and properties. Polysaccharides, i.e., high-molecular weight carbohydrate biomolecules, such as cellulose, chitin, etc., are widely distributed in nature, biocompatible, biodegradable and have a set of promising properties. Polysaccharides and biomass were one of the first class of polymers studied during the expansion of mechanochemistry technology to organic substances. The polysaccharides gain a lot of attention among recent works on mechanochemical modification of organic substances as well [8,9]. This systematic review belongs to the topical collection focusing on “State-of-the-Art Macromolecules in Russia” and describes evolution and results of work on mechanochemical modification of polysaccharides in the Russian Federation, but all the discussed trends are part of the mainstream of mechanochemistry research worldwide.

### 1.2. Aims

The aim of this study is to systematically review the trends and results of research works on modification of polysaccharides under mechanochemical conditions. The evolution of research articles on mechanochemical modification of polysaccharides starting from earlier works in the 1960s to recent days in the Russian Federation is described and analyzed in terms of world-wide trends.

## 2. Methods

The systematic review was carried out to track publications on mechanochemical modification of polysaccharides across the subject area and apparatus used to perform the substance treatment. This systematic review was conducted by following the reporting checklist of the Preferred Reporting Items for Systematic Reviews and Meta-Analyses (PRISMA). The search strategy incorporated examination of electronic database Scopus, eLIBRARY and in some cases the websites of journals and libraries to find the full-text of earlier documents. The keywords used in the search included the following: “(Mechanochemistry OR Mechanochemical) AND Polysaccharide” while no date and language restrictions were imposed. The country/territory filter “Russian Federation” was used. Due to some of the papers referring only to a specific method of mechanochemical activation (ball milling, solvent-free extrusion, etc.) a manual search of the reference lists within the found articles was carried out to collect additional relevant publications. All authors of this review independently analyzed the obtained papers to exclude ones with incomplete data or that were out of scope. The latter means that polysaccharide or mechanochemistry were just mentioned within the article, and the main topic of the paper was irrelevant to mechanochemical treatment or modification of polysaccharides. Two authors extracted the data from the included papers and the other author checked the extracted data. All reviewers worked with the obtained papers with the aid of reference management software, compared their decisions and resolved differences by consensus. The last search was run on 3 June 2022.

The first literature screening of records referring to “Polysaccharide AND (Mechanochemistry OR Mechanochemical)” showed more than three thousand research records (Figure 1). The application of the country/territory filter “Russian Federation” revealed 131 documents referring to keywords “Mechanochemistry AND Polysaccharide” and 201 documents dealing with “Mechanochemical AND Polysaccharide”. A screening of these records for duplicates, documents with incomplete data or that were out of scope gave us 79 relevant documents. The earliest found full-text record dealt with mechanochemical treatment of cellolignin in the presence of sulfuric acid using a vibration mill [10]. A hand search of the reference lists within the found records showed that a lot of papers describe results of mechanochemical modification using more specific terms, such as ball milling, shear deformation, solvent-free reactive blending, solvent-phase mechanical activation, etc. These documents were out of the first literature screening and had to be added. The manual search revealed an additional 29 documents. 

### Risk of Bias

Current analysis of the data extracted from studies written at the end of the twentieth century could introduce a significant bias due to incomplete data. Numerous limitations have arisen from non-referral to mechanochemistry as to the method of polysaccharide treatment. A manual search of the reference lists within the filtered papers showed that such types of treatment were identified as ”grinding”, “under conditions of shear deformation in solid state”, “solid-phase mechanical activation” etc.

## 3. Results and Discussion

### 3.1. General Trends

The earlier works on mechanochemical modification of the polysaccharides were focused on destruction of polysaccharides under mechanical stress [10]. Chemical reactions of the generated free radicals were carefully studied using EPR spectroscopy [11]. Such types of works contained references, which could be tracked to earlier works about ball-milling of woods (i.e., cellulose) by P.Y. Butyagin [12] or other polymers by N.K. Baramboim [13] from the 1960s. Of course, the history of mechanochemistry starts much earlier. For example, American physicist P.W. Bridgman, who received the Nobel Prize in physics in 1946, worked on the effect of high pressure and shear deformations on stability of a wide range of substances [14].

Further works on mechanochemical treatment of polysaccharides extended the range of polymers and focused mainly on:(1)mechanical activation of natural polysaccharide-containing sources to enhance extraction of components;(2)mechanochemically assisted immobilization of non-/poorly-soluble bioactive compounds onto water-soluble polysaccharides in order to enhance drug solubility and bioavailability;(3)mechanosynthesis of polysaccharide derivatives and copolymers and fabrication based on materials.

Figure 2 shows the main trends of mechanochemical activation of polysaccharide-containing substances, and the apparatus used for the treatment.

### 3.2. Ball Milling

A majority of works are still done on mill-type machines. For example, works on the fabrication of organic and hybrid composite materials are systematically carried out at the Institute of Solid State Chemistry and Mechanochemistry of the Siberian Branch of the Russian Academy of Sciences under the supervision of Academic N.Z. Lyakhov [15,16]. A series of works on the modification of polysaccharides and low-molecular weight biologically active substances using milling apparatuses of various types, were carried out there by A.V. Dushkin with colleagues from the Research Institutes of the Siberian Branch of the Russian Academy of Sciences [17,18]. The phenomenon of aggregation of solid reagents during mechanical activation of their mixtures was discovered and studied. The ability of reagents to aggregate, according to the authors, determines the possibility of their further interaction both during mechanical treatment and after it. The result of the interaction is an increase in solubility and in the effectiveness of the pharmacological action of the drugs. The authors found that, under the joint mechanical treatment of various drugs and polysaccharides (arabinogalactan, pectin, hydroxyethylstarch, carboxymethylcellulose, dextran) or oligosaccharides (β-cyclodextrin, glycyrrhizic acid, etc.) the water-soluble complexes of a “host–guest” type, including micelle-like dispersions [19,20,21,22,23,24,25,26,27,28,29,30], or the solid dispersions in which biologically active substances are in an amorphous state or distributed in a molecular level [31,32,33,34,35] can be obtained. Pharmacological tests have shown that the mechanochemical treatment of the solid dispersions of drugs in the polysaccharide matrices increased their bioactivity by 2–10 times (for example, antibiotics and antioxidants) while reducing the effective dose more than 2–30 times keeping the basic pharmacological activity; decrease in the incidence of side effects also was marked [36,37,38].

#### 3.2.1. Drug Delivery Systems

The mechanical treatment of the solid polymer mixtures can be used to prepare a non-covalent polymer hydrogel. In particular, the results obtained from the thermodynamics study, particle size analysis, and electron microscopy show that chitosan and sodium alginate form a pH-sensitive hydrogel and also maintain good stability in aqueous solution [39]. In vivo study showed that the pH-sensitive hydrogel could achieve the targeted release of the mechanochemically loaded 5-amino salicylic acid with a retention time over 12 h. The authors noted that this approach could allow production of efficient colon-targeted drug delivery systems.

Preparation and study of the biological activity of mechanocomposites of piroxicam and meloxicam with organic and inorganic carriers, including chitosan and microcrystalline cellulose, undertaken at the Institute of Solid State Chemistry and Mechanochemistry of the Siberian Branch of the Russian Academy of Sciences in order to create new dosage forms was described in the series of works by T.P. Shachtschneider et al. [40,41,42]. The results showed that mechanical activation in a ball mill leads to the formation of stable molecular complexes due to hydrogen bonds, which allowed changing the solubility and dissolution rate of the initially crystalline and poorly soluble drugs. Ball milling of betulin dipropionate with the water-soluble polysaccharide arabinogalactan, allowed an increase in the dissolution rate in comparison with the initial botulin derivative due to disordering of crystal structures of the biologically active substance, and the formation of molecular complexes with the polymer. The studied complexes showed a targeted antitumor effect, causing apoptosis of cancer cells without affecting healthy cells. Apparently, the mechanism of action of the samples on cancer cells was similar to betulinic acid and affected mitochondrial function, increasing their permeability to reactive oxygen species.

Mechanocomposites of the antiviral drug rimantadine and cyclophosphamide with arabinogalactan or its oligomers, were studied by V.A. Babkin with co-authors at the Favorsky Irkutsk Institute of Chemistry of the Siberian Branch of the Russian Academy of Sciences [43,44,45]. According to IR and ^13^C, ^15^N and ^31^P NMR spectroscopy, the drugs formed hydrogen bonds with the polymer after a long-term mechanochemical ball milling (>4 h). The disordering of drug crystalline structure occurs in the matrix of the polysaccharide after treatment of solid mixtures in the roller mill for 4–12 h. On the other hand, no changes in the structure of the natural biologically active compound arglabin under “soft” conditions of mechanochemical treatment (roller mill with a ball diameter of 9 mm, treatment duration 20 h) were found. All spectral data (IR, UV, ^13^C NMR, WAXD) were identical to those for the original terpenoid, and the authors did not reveal an increase in its solubility and bioavailability.

#### 3.2.2. Plant Protection Systems

A number of plant protection products were obtained through ball milling [46,47]. Improvement of pesticide penetration into seeds has been detected for arabinogalactan nanocomposites: Five-fold for tebuconazole and imidacloprid, and more than ten-fold for imazalil and prochloraz. The authors assume that the effect of poly-/oligosaccharides on penetration might be associated with the solubility enhancement and the improved affinity of the delivery systems to the surface of grains [48]. The results of efficiency estimation of spring wheat varieties treated with a supramolecular complex of tebuconazole with arabinogalactan, prepared according to the original mechanochemical technology, demonstrated that a single treatment of crops at the beginning phase of earing can effectively control leaf infections of soft spring wheat [48]. Mechanical treatment of poorly soluble pesticides (chlorsulfuron, carbendasim, and thiram) in the presence of arabinogalactan found that the polysaccharide undergoes polymer chain breakage, splitting of the side fragments, and the appearance of new functional (carboxyl) groups. The shear deformation of pesticides leads to disordering of their crystalline structure with breakage of the intermolecular hydrogen bonds that results in increases in the pesticide solubility (from 1.5 to 5.0 times). The formation of the inclusion complexes with the participation of the nitrogen-containing groups of the herbicide and hydroxyl and ester groups of the polymer was confirmed using NMR, UV and IR spectroscopic methods. It was shown by the authors that usage of such compositions allowed decreases of pesticide consumption by ten-fold [49,50].

#### 3.2.3. Mechanical Activation of Drugs in Mixed Aqueous Solutions

In addition to the considered works devoted to increasing the specific activity of drugs by processing solid dispersions of reagents in milling apparatuses of various types in the presence of polysaccharides, a number of works performed in aqueous media using mechanochemical technologies should also be noted. Thus, a team of authors from the Research Institutes of the Siberian Branch of the Russian Academy of Sciences has shown that microwave heating is a highly efficient technique to prepare the supramolecular complex of betulin diacetate (isolated from birch bark) and the natural polysaccharide arabinogalactan [51]. The reaction time was reduced from several hours to a few minutes in comparison with the traditional procedure. A change in the size and surface morphology of drug crystals under microwave heating was observed, suggesting that the microwave impact facilitates betulin dissolution in aqueous media that could contribute to high-speed synthesis of the supramolecular complex. The obtained complex exhibited antitumor activity against Ehrlich ascites carcinoma cells comparable to the system obtained in the conventional way. A novel environmentally friendly process of introducing a hardly soluble rutin into chitosan edible films under short-term (10 s) mechanical activation of mixed aqueous solutions in a metastable state using a rotor–stator device was proposed [52]. It was shown that mechanical activation accelerates the nucleation of rutin and prevents aggregation of the resulting particles due to them embedding in the polymer structure. It was found that the rutin particles’ size decreases with an increase in its concentration in supersaturated solutions, as well as with an increase in the chitosan solution viscosity. Rutin particles had a needle-like shape with a diameter of 0.40–0.75 μm and showed a reinforcing effect, increasing the tensile strength of chitosan films by 25–40%. Chitosan films containing rutin were smooth, uniformly yellow colored, and had lower water vapor permeability compared to films made of chitosan alone.

#### 3.2.4. Study of Changes in the Structure of Polysaccharides after Ball Milling

The authors also pay attention to changes in the structure and properties of the polysaccharides themselves during mechanochemical processing. “Mild” conditions for mechanical treatment in the rotary (roller) mill do not lead to destruction of polysaccharide macromolecules, as was assessed by gel permeation chromatography [53]. More intense processing regimes (planetary mill and rotary mill with mixed ball loading) led to significant decreases in the molecular weight of arabinogalactan macromolecules (about twice). Changes in the molecular mass distribution, monosaccharide composition, and the degree of branching of its macromolecules due to the partial destruction of polysaccharide macromolecules and subsequent recombination of resulting fragments were found [54]. Toxicological studies of the ground arabinogalactan, hydroxyethyl starch, fibregum, and dextrans showed that they can be classified as low-toxicity substances [53]. The destruction of amylose at high pressures (20 MPa) combined with shear (540°) was studied using a Bridgman anvil. According to mass spectrometry data, the formation of an appreciable number of low-molecular oligomers was found and explained by the predominant cleavage of chains near the boundary zones of crystallites. Free radicals appeared in the treated samples and were recorded by EPR spectroscopy. After the mechanochemical treatment, the samples became colored from yellow (with wavelengths from 565 to 590 nm) to brown tints that the authors attributed to the dehydration of the macromolecular radicals and the appearance of double bonds in their framework [55]. The structural, morphological and chemical transformations in α-chitin caused by mechanical activation of its aqueous suspension in the rotor–stator device for 5–30 s were studied using optical and electron scanning microscopy, N_2_ adsorption/desorption isotherms, X-ray diffraction and FTIR-spectroscopy [56]. It was established by the authors that the specific surface area, porosity and the deacetylation degree of chitin increase with increasing processing time. Thus, activated chitin demonstrated a limiting adsorption capacity to adsorb anionic and cationic dyes on average two to four times higher, and three to four times less time to reach equilibrium as compared to raw chitin.

#### 3.2.5. Biomass Mechanical Transformations

A separate series of investigations devoted to the study of the impact of mechanochemical processing of lignocellulose raw materials in various mills was carried out by O.I. Lomovsky together with colleagues from the Research Institutes of the Siberian Branch of the Russian Academy of Sciences [57]. The authors note that the activation of the mixtures of solid components (an increase in the interface area, accumulation of defects, amorphization, and increase in free energy) can create unusual conditions for chemical reactions, with temperature and pressure arising during mechanical treatment. The effect of mechanical treatment including alkaline and enzymatic conversions on changes in the composition and antioxidant activity of the polysaccharide fractions of the moss Sphagnum fuscum, and sphagnum moss peat was studied [58]. The increased antioxidant activity of the peat preparations was shown due to an increase in the concentration of the hydroxyphenol compounds after mechanochemical treatment. The catalytic activity of the polysaccharide and humic fractions extracted from mechanically activated peat, have also been discovered [59]. In general, the conducted studies show that mechanical and mechanochemical treatment of plant mass of different origin can significantly increase the yield of water-soluble fractions and modify their beneficial properties, due to an increase in the concentration of functional groups and the appearance of new ones [60,61,62,63,64]. The significance of the interphase processes, changes in surface chemistry, related dimensional effects, and the disordering of the crystal structure and amorphization were demonstrated [65].

#### 3.2.6. Mechanochemical Synthesis of Nanocatalysts

In a number of works mechanochemical milling has been positioned as a very attractive nanocatalysts synthesis technique using polysaccharides as sacrificial templating agents, those which could be obtained from agricultural wastes through microbial fermentation. In order to develop new economic and efficient catalysts and to minimize the impact on the environment according to green chemistry principles, the systems based on niobium oxide (low-cost chemical) and polysaccharides (renewable, sustainable and low-cost production) were synthesized through mechanochemical processes [66]. The authors note that mechanochemical milling provides a powerful technique for minimizing the use of hazardous solvents. Mechanochemical modification of starting polysaccharide was confirmed by IR and X-ray diffraction patterns and showed to affect the crystallization of niobium during calcination step. Synthesized nanomaterials exhibited low porosity (generally interparticular meso- and macroporosity) as well as low acidity values (both Brönsted and Lewis related measurements). Despite their low acidity as has been compared to most reported solid acids in literature, the synthesized nanocomposites exhibited a remarkable catalytic activity during the selective oxidation of isoeugenol to vanillin under conventional heating (90 °C). The Nb/ZnO nanocatalysts were also synthesized by a mechanochemical-assisted and sacrificial template method [67]. Their catalytic properties were optimized for the efficient conversion of levulinic acid obtained from lignocellulosic biomass to N-heterocycles under mild reaction conditions and the absence of solvent. High conversion and selectivity values up to 94.5% and 97.4% were achieved by employing zinc oxide nanocomposite modified with 10% of niobium. The high utilization of biomass derived feedstocks, combined with environmentally friendly procedures and heterogeneously catalyzed processes, could help moving on from the current fossil-fuel dependent industry to more sustainable bio-based ones.

### 3.3. Reactive Extrusion

The study of the kinetics of solid-state reactions is difficult due to the mismatch between the residence time of the substance in the reactor and the duration of the chemical reaction, and for them it is impossible to indicate general kinetic laws, as in the case of homogeneous reactions in the liquid or gaseous phase. Therefore, in the field of the theory of mechanochemical synthesis, a macrokinetic approach is used, which considers the process in the volume of the entire reactor and takes into account, as the main factors responsible for the rate of the mechanochemical reaction, activation of reagents, changes in the properties of the reaction surface, as well as the contribution of temperature to the increase in energy systems and dynamics of relaxation processes [68]. The key factor determining the course of the reaction is not the time of deformation, but the energy required to overcome the elastic limit of the solid, i.e., to transfer the system to a state of plastic flow. In this state the absolute diffusion restrictions arising due to the aggregate state of the reagents are removed, and an intense accumulation of energy is observed with the formation of spot and linear (dislocations) defects, which make it possible to reduce the activation energy of the subsequent chemical transformation of the substance. The intensity and direction of the reaction, in turn, depend on the amount of accumulated energy and on the specific properties of the substances involved in the reaction. A high level of mechanical activation for organic systems is impossible due to low thermal stability and long times of relaxation processes of organic compounds. As a result, the greatest progress in the study of organic mechanochemical reactions has been achieved on devices that perform continuous deformation and more precise temperature control: Bridgman anvils, twin-screw extruders, and specially designed planetary mills.

Reactive extrusion is one of the most popular and important areas of research to create new materials in the world. However, its use is limited by synthetic polymers capable of melting without side destructive processes (polyolefins, polyamides, polyesters), and thermoplastic natural polymers (mainly starch), which are pastes with a high content of water and other plasticizers, and the ability to create conditions for effective mechanochemical action on reactive systems in melts or pastes (gels). Works devoted to the study of such processes indicate that the chemical interaction of components in such systems is very insignificant. Due to the high energy of cohesion, the melting temperature of polysaccharides is higher than the temperature of their decomposition, and when creating polymer composites by traditional melt methods of reactive extrusion, native polysaccharides serve mostly as fillers [69].

#### 3.3.1. Reactive Blending of Polysaccharides with Thermoplastic Polymers in the Molten State

In the work carried out at the Department of Polymers and Composite Materials of the Semenov Federal Research Center for Chemical Physics of the Russian Academy of Sciences by S.Z. Rogovina et al., blends of various thermoplastic polymers with polysaccharides (cellulose and derivatives, starch, chitin, arabinogalactan) with improved biodegradability and a number of functional composite materials based on chitosan were obtained in a melt of the thermoplastic components under shear deformation [70,71]. The blends of polylactide with starch of various ratios were obtained in a closed type mixer (Brabender). The thermal behavior of the obtained composites was studied by DSC, and their crystallinity was determined using X-ray diffraction; according to GPC data, the biodegradation of these composites upon exposure in the soil was quantified. An analysis of the experimental results shows that polylactide is almost incompatible with starch. A decrease in the melting point of polylactide in the composites with an increase in the starch content indicates the formation of more defective small crystals due to the difficulty of the crystallization process. On the basis of the data of X-ray structural analysis, it was shown that the total degree of crystallinity of the composites increases after exposure in the soil. According to the authors, this is associated with the washing out of the amorphous polylactide region under the action of water and microorganisms in the soil. Using GPC, it was revealed that the biodegradation of polylactide when exposed in the soil proceeds according to a depolymerization mechanism; in the composites, this process occurs more intensely [72]. Crumb rubber compositions with starch and cellulose were obtained in a Brabender mixer. It has been shown that the mechanical characteristics of both compositions differ only slightly from each other. When the content of polysaccharides in the mixture increases, the elastic modulus grows and the strength and elongation at break reduce. The addition of polyethylene glycol slightly affects the values of the mechanical parameters. After exposure of the samples in soil, an increase in mechanical characteristics, which was likely to be due to the occurrence of crosslinking of crumb rubber by radicals formed under the influence of the environment, was observed [73]. Orevac–crumb rubber–arabinogalactan blends of various compositions were obtained in a Brabender mixer. The structure, mechanical properties, and biodegradability of the materials were studied. It was shown that an increase in arabinogalactan content leads to an increase in elastic modulus and a decrease in elongation at break, as well as slightly affecting the tensile strength. The mass loss of the samples after exposure in soil defining their biodegradability rises with an increase in arabinogalactan content in the composites [74]. The ternary powder blends based on low-density polyethylene, poly (lactic acid) and starch were obtained in a rotor disperser at 130 °C [75]. The apparatus is based on a single-screw extruder (the screw diameter of 32 mm, the length to diameter ratio is equal to 11, and the rotation velocity of 45 rpm). To create the intense shear strains the rotor disperser has a grinding head designed as a cam element rotating inside a channeled cylinder and equipped with a cooling jacket. The joint action of temperature and shear deformations resulted in formation of the polymer powders. It was found that the presence of two polymers of natural origin in the system with a total mass fraction of 60% promotes intensive biodegradation studied via imitating the environmental conditions and tests on fungal resistance.

#### 3.3.2. Solid-State Reactive Blending of Polysaccharides

Differential scanning calorimetry was used to establish that, at 20–200 °C, an endothermic process proceeds in polysaccharides (cellulose, methyl cellulose, cellulose di- and triacetate, chitin, chitosan, starch, etc.) whose enthalpy is noticeably enhanced after plastic deformation of the polymers at 0.5–2.0 GPa using the Bridgman anvil. By the example of cellulose, it was shown that the process was associated with water present in the polymers. An increase in the enthalpy as a result of pressure treatment was due to both structural transformation of cellulose I into cellulose II and the intensification of intermolecular interactions in the polymers that was due to the formation of a system of electric charges in the samples [76]. Microcrystalline cellulose, starch, methyl cellulose, chitin, and their mixtures with polyacrylamide have also been studied by differential scanning calorimetry and thermogravimetry. The thermograms in the temperature range of 20–200 °C display endothermic peaks describing the process of destruction of hydrogen bonds in the polysaccharides. The endothermic process was accompanied by mass loss associated with desorption of the adsorbed water which played a major role in the formation of hydrogen bonds in the polymers. As a result of the treatment using an anvil type high-pressure device, the enthalpies of endothermic processes in individual polymers increased by a factor of 2.8 but decreased in the polymer blends. In the processed blends of microcrystalline cellulose with polyacrylamide, up to 36% of cellulose became water-soluble [77].

Thus, solid-state reactive blending is one of the most promising and effective approaches to obtaining various macromolecular systems based on synthetic and natural polymers that are not miscible under normal conditions. The impact of pressure and shear strains on solid reagents leads to the formation of active states in the bulk of a substance, the disordering of its structure and the polarization of molecules [78], that results, in particular, in the intense polymerization of solid vinyl and cyclic monomers [79], ultra-high mobility of terminal functional groups and the ability of heterochain bonds of synthetic polymers to direct reactions with functional groups of polysaccharides [80,81].

A mechanochemical approach for forced dispersion of solid reagents during reactive extrusion was successfully developed by a team of authors at the Institute of Synthetic Polymeric Materials of the Russian Academy of Sciences to develop environmentally safe processes for the chemical modification of polysaccharides with a significant reduction in the synthesis duration and consumption of reagents. The method is also suitable for obtaining hybrid nanocomposites based on polymers and functional fillers immiscible under usual conditions due to mechanically stimulated chemical interaction in their mixtures. Mechanical activation of all considered systems is carried out in the absence of solvents and diluents; the studied synthesis temperature range lies below the softening (melting) temperature of all reagents.

The main research results were obtained using a twin-screw extruder from Berstorff (Hannover, Germany), an efficient co-rotating mixer with 40 mm in screw diameter, specially equipped for the deformation of high-viscosity reactive systems, including polymer mixtures and solid dispersions. The working elements of the screws provide deformation mixing of the material with the ability to control the temperature in the working areas in the range from −5 to 300 °C and the intensity of mechanical action by varying the sets of screw elements and their rotation speed. Such equipment ensures both the production of pilot batches of samples and the full scale-up of processes for the creation of solvent-free technologies for the production of materials based on biopolymers.

##### Wood Delignification and Cellulose Mercerization

The study of the impact of extrusion processing of solid polysaccharides and polysaccharide-containing raw materials, began under the supervision of Academician N.S. Enikolopov from the study of the processes of wood delignification and cellulose mercerization [82], for which the corresponding patents of the Russian Federation were obtained. It was found that the hydrolytic processes of birch wood in the presence of catalysts for polysaccharide hydrolytic cleavage (acids, alkalis, and enzymes) are accelerated by several orders of magnitude in the extruder force fields while the working temperature was considerably reduced (relative to thermohydrolysis) [83]. The authors claim that the use of a mechanochemical approach allows prevention of the formation of harmful side products (furanoids), relative to their closest analog the Kraft process for producing cellulose from wood.

##### Chitin Deacetylation

Further, it was shown that alkaline treatment of solid mixtures of chitin with alkali in an extruder can produce highly deacetylated chitosan (up to 0.98) with a low degree of polydispersity of macromolecules, which is important in the development of new promising materials in biotechnology and biomedicine [80,84]. Comparison of the viscosity behavior, IR spectra, and MMR of chitosan synthesized by deacetylation in the solid state with samples obtained by the industrial suspension method showed the identity of their molecular structure and, consequently, the homogeneity of chemical reactions in the bulk of solid reagents during their joint deformation. At the same time, the processing time of chitin was reduced from hours to minutes, and the alkali consumption was half as much (NaOH: chitin weight ratio was not higher than 1). X-ray diffraction studies were performed to investigate the structural changes in chitin under pressure and shear during its solid-state processing using a twin-screw extruder and Bridgman anvil; the structure of the obtained chitosan was studied as well [85]. Deformation under the conditions of solvent-free grinding at room temperature reduced the crystallinity of the original chitin. Addition of water restores the crystallinity of the material up to the value characteristic of the original chitin. Extrusion of chitin at room temperature with addition of water preserves the crystallinity and degree of ordering of the chitin crystal lattice, the same as solvent-free grinding at an elevated temperature (180 °C). The maximum degree of amorphization of chitin is attained by its processing on a Bridgman anvil. Solid-state synthesis of chitosan from chitin leads to a product with a more amorphous structure in comparison with chitosan produced by the suspension method.

##### Chitosan Derivatives

Interest in the modification of chitosan as one of the polysaccharides is associated with the possibility of practical applications of chitosan and its derivatives in medicine, biology, and pharmaceuticals. Chitosan derivatives, in particular, are used as carriers in the preparation of wound dressings, porous composite materials for filling bone defects, implants, cell culture matrices, biodegradable film materials, as well as materials used in chromatography for the separation and purification of biologically active compounds (nucleic acid acids, steroids, amino acids). Due to the film- and fiber-forming abilities, sorption and complexing properties, chitosan and its derivatives are widely used in the creation of ion-exchange, dialysis, and ultrafiltration membranes. One of the most important directions in the use of chitosan is the creation of prolonged pharmacological agents as a carrier of drugs and other biologically active molecules. Due to the sensitivity of preparations based on chitosan and its derivatives to the acidity of the medium, problems arise in achieving the required solubility of preparations based on chitosan in neutral media, which is fundamentally possible when water-soluble polymers are grafted onto polysaccharide macromolecules. However, due to the fact that chitosan dissolves only in acidic aqueous media and is completely insoluble in organic solvents, the possibilities of liquid-phase modification of chitosan with water- and organic-soluble substances, in particular, monomeric and polymeric reagents, are limited. For this reason, it is natural to be interested in the possibilities of solid-state processes of chemical modification of chitosan.

New graft copolymers and chitosan derivatives which possess atypical solubility in neutral aqueous or organic media compared to the original polymer, have been obtained by solid-state reactive mixing in an extruder. Such modification radically changes the possibilities of forming materials using innovative technologies for polysaccharides. In the solid-state preparation of acylated and carboxymethylated derivatives of chitin, chitosan, and cellulose in the extruder [86,87] as well as mixtures of cellulose with chitin (chitosan), including those modified with a crosslinking agent [88], high degrees of conversion of reagents were achieved. It has been shown that during the formation of salts of solid organic acids and chitosan, the degree of protonation of chitosan correlates with the strength of the acid, and therefore, there are no diffusion restrictions for this fast reaction in the solid state. The reactions of N-, O-acylation and carboxymethylation of chitosan were sensitive to the ratio of reagents, temperature of the treatment, and the comparative reactivity of the functional groups of the polymer, that is, they were also chemically controlled in the solid state of the reagents during deformation. Comparison of the average viscosity molecular weight and particle size distribution of chitosan obtained by deacetylation in an extruder with those for industrial chitosan subjected to similar mechanical treatment showed that the contribution of mechanical degradation to the drop in the molecular weight of the polymer is insignificant, especially at elevated temperatures. Therefore, the decrease in the polymer molecular weight with the reactions of deacetylation and carboxymethylation in the solid state is caused mainly by the process of thermal oxidation of polysaccharides in the presence of strong bases.

The solvent-free synthesis of allyl-substituted chitosan derivatives through reactive co-extrusion of chitosan powder with allyl bromide at shear deformation was performed [89]. The total content of allyl substituents from 5 to 50 per 100 chitosan units as a function of the component ratio in the reactive mixtures was revealed. Carrying out the reaction without any additives leads to the selective formation of N-alkylated derivatives, whereas in the presence of alkali the ethers of chitosan were preferentially formed. The results suggest that the proposed approach allows significantly higher yield of products to be obtained at high process speeds and significantly lower reagent consumption, as compared with the liquid-phase synthesis in organic medium. The synthesized unsaturated derivatives were used as photosensitive components in laser stereolithography for fabrication of three-dimensional biocompatible structures with well-defined architecture [90]. The mechanochemical reactions of chitosan alkylation when interacting with docosylglycidyl and hexadecylglycidyl ethers in the absence of solvents at shear deformation in a pilot twin-screw extruder also were studied [91]. According to calculations for products soluble in aqueous media, it was possible to introduce about 5–12 hydrophobic fragments per chitosan macromolecule, with a degree of polymerization of 500–2000. The length of the carbon chain of the alkyl substituent significantly affects its reactivity under the chosen conditions of mechanochemical synthesis. It was shown that modification disturbs the packing ability of the macromolecules, resulting in an increase in plasticity and drop in the elastic modulus of the film made from the hydrophobically modified chitosan. These derivatives could be used as rheological modifiers, selective sorbents, and stabilizers for compositions intended for various applications.

##### Chitosan Graft Copolymers and Multicomponent Composites

The processes of grafting onto chitosan fragments of water-soluble natural and synthetic polymers, polyvinyl alcohol and hydroxyethyl cellulose, occur due to the reactions of their terminal aldehyde groups and amino groups of chitosan under mechanochemical action in a twin-screw extruder [80,92]. A possible mechanism is considered, which consists of the intermediate formation of terminal aldehyde groups of polyvinyl alcohol and their further activation, leading to interaction with chitosan amino groups during plastic deformation of polymer mixtures. The ability of copolymers to act as stabilizers of dilute and concentrated aqueous dispersions of inorganic nanoparticles and biologically active substances was found, in contrast to mixed systems of the same composition that do not exhibit such properties [93]. Composite systems, which form stable suspensions in an acidic aqueous medium and are capable of forming micro-heterogeneous materials during melt casting, have been obtained through solid-state interaction of maleic anhydride, chitosan, and polyethylene in the extruder. A model reaction of solid-state modification of chitosan with maleic anhydride has been carried out, reaction products have been identified, and a possible mechanism for the grafting of polyethylene to chitosan through the intermediate formation of chitosan and maleic anhydride derivatives has been proposed [94]. A model reaction of solid-state graft polymerization of lactide onto chitosan was studied and its mechanism was established. The high activity of chitosan amino groups in solid-state reactions of nucleophilic addition to the lactide ring and the ability of dilactone to graft polymerization due to the addition of the substituent hydroxyl group formed during ring opening to the energetically strained monomer ring were found. As a result, copolymers of chitosan and the corresponding oligomers were obtained with an average degree of polymerization of monomers in the grafted chains of three to ten, and a grafting degree of up to 160%, which have amphiphilic properties with a tendency to disperse in organic solvents at the colloidal level. It has been established that the dispersed phase is a set of micellar-like structures of the “core-shell” type, consisting of polysaccharide aggregates stabilized in a hydrophobic medium by grafted polyester chains. Chitosan-g-lactide copolymers were obtained as well by co-reactive extrusion of chitosan with aliphatic polyesters. A mechanism of grafting due to aminolysis reactions of ester bonds of polymers and reactions of their terminal carboxyl groups with amino groups of chitosan was proposed and substantiated [95]. Data have been obtained on relaxation transitions, crystallization dynamics, and mechanical properties of such copolymers. It has been shown that these graft copolymers have high mechanical characteristics when obtaining films from the melt of samples containing up to 40 wt.% polysaccharide. It has been established that the modification preserves the valuable biochemical properties of chitosan, which make it possible to improve the sorption capacity and biocompatibility of materials and give them the ability of controlled biodegradation. Natural proteins can be included in the composition of materials based on chitosan and polylactide directly during the copolymerization process without decomposition (in particular, collagen and gelatin). It has been shown that formation of complexes of chitosan with proteins proceeds with a quantitative yield already at room temperature during deformation under pressure in the solid state. Using the trypsin enzyme as an example, the authors showed that during plastic deformation under pressure of solid mixtures containing chitosan and the enzyme, the latter interacts with the polysaccharide at the molecular level, retaining its biocatalytic activity when released from the matrix into solution [96]. The enzyme immobilized in this way has a pronounced prolonged action. This opens up new possibilities and expands the experimental base for solving both fundamental and applied problems of creating dosage forms with immobilized enzymes.

The synthesized derivatives, copolymers and multicomponent composites were used for fabrication of a wide range of materials for biomedical application in the form of films, non-woven mats, porous hydrogels and microparticles [97,98,99]. Biocompatibility and biodegradability of the materials based on modified in solid state polymers were proved in a course of in vitro and in vivo experiments. Using the method of solid-state reactive extrusion, formulations for boron-neutron capture therapy were created [100]. The polycomplex hyaluronic acid: boron obtained by solid-state synthesis is not inferior in stability to covalently bonded systems, since it contains polychelate fragments distributed in many of the chains in the macrocomplex. The simplicity of synthesis, the huge advantage in the yield of the initial ^10^B isotope (tens or even hundreds of times), makes this approach very promising. The complexes are stable both in the form of aqueous solutions and when administered in vivo; the drug is non-toxic.

Thus, the possibility of carrying out solid-state synthesis of derivatives and graft copolymers of polysaccharides during mechanical activation in a pilot twin-screw extruder has been shown. As a result, new materials based on biopolymers have been obtained, the synthesis of which is complicated or impossible using solution- and melt-based technologies. It has been established that joint plastic deformation of the studied solid mixtures containing chitosan and organic reagents at temperatures below their softening or melting temperatures (mainly in the range of 0.5–0.7 melting temperature) leads to dispersion of the mixture components at the segmental-molecular level and is accompanied by their interaction with the formation of hydrogen, ionic and covalent bonds. This is consistent with existing ideas in the field of the theory of mechanochemical synthesis on the preference of low temperatures for the efficient occurrence of solid-state chemical reactions. Overcoming diffusion limitations due to the applied mechanical action under the selected process conditions leads to a high degree of conversion of the functional groups of the reagents. The process conditions: pressure, magnitude of applied shear strain, treatment time, and ratio of components have a significant effect on the conversion and are selected individually for each reactive system depending on its mechanical properties. It was found that the product yields are maximum at content of polysaccharide over reactive mixtures within 40–60 wt.%. Table 1 summarized the products of solid-state modification of polysaccharides and their possible applications.

## 4. Conclusions

This systematic review of the literature has shown that the interest in studies of processes occurring under mechanochemical impact on polysaccharides, attracts the constant interest of researchers in various fields of science. Compared with solid-state reactions of inorganic substances and low molecular weight organic compounds, the mechanisms of solid-state transformations of polysaccharides are still poorly understood. At the same time, mechanochemical approaches seem to be the most acceptable method for solving the problem of thermodynamic incompatibility of natural and synthetic substances. Works in this direction are characterized by an interdisciplinary approach, involving fundamental complex research at the junction of several areas, such as the synthesis of amphiphilic macromolecules and the study of chemical processes occurring in polymer matrices during their structuring (polymer chemistry); development of hybrid biomimetic systems for solving biomedical problems (materials sciences); development of methods for structuring polymer systems using additive technologies (physics); determination of biocompatibility, bioavailability and cytotoxicity parameters of obtained materials (biomedicine) and unites specialists working in these fields. The systematic study of such processes is an urgent task not only from the point of view of fundamental science, but also for finding and developing opportunities for creating new functional polymeric materials using efficient and environmentally friendly methods.

## Figures and Tables

**Figure 1 ijms-23-10458-f001:**
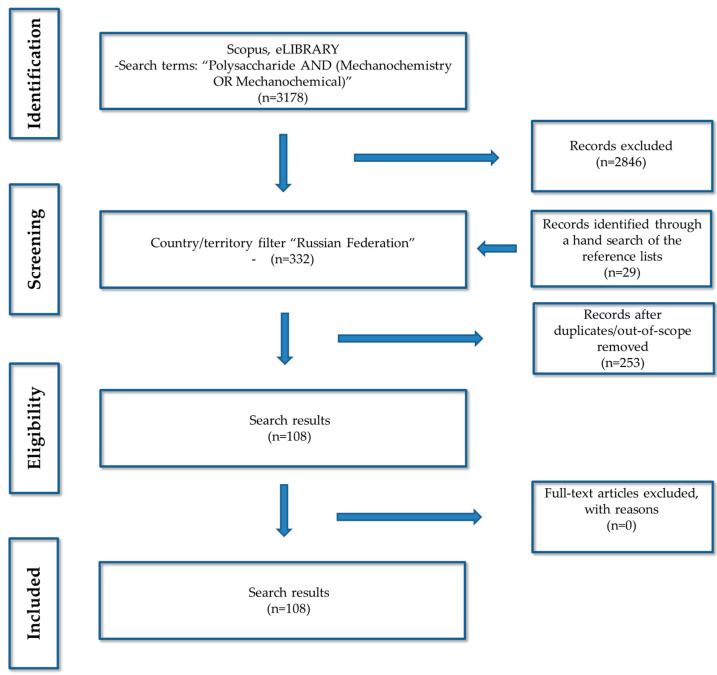
The PRISMA flow diagram.

**Figure 2 ijms-23-10458-f002:**
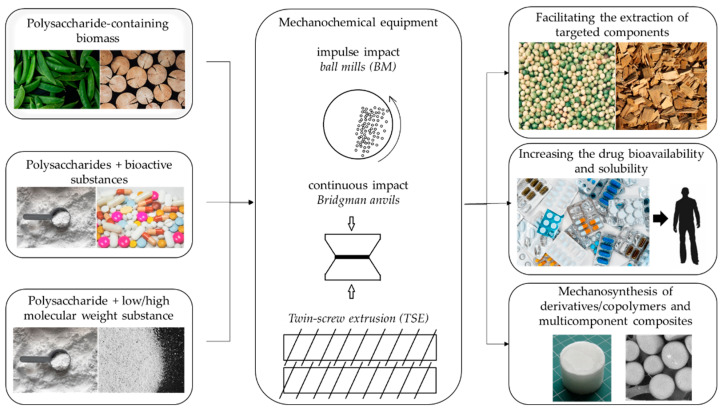
The main trends of mechanochemical modification of polysaccharides.

**Table 1 ijms-23-10458-t001:** Components and devices used for solid-state mechanochemical modification of polysaccharides tailored for various applications.

Polysaccharide	Additives	Mode of Action ^1^	Products	Applications	Refs.
Chitosan/Sodium Alginate	5-amino salicylic acid	BM	pH-sensitive hydrogel	Drug delivery	[39]
Chitosan, microcrystalline cellulose	Piroxicam; meloxicam	BM	Complexes due to hydrogen bonds (H-bonds)	Drug delivery	[40,41]
Arabinogalactan	Sibazon, Mezapam, Azaleptin, and Indometacin	BM	H-bonds	Drug delivery	[19]
Arabinogalactan	Carotenoids (β-carotene, canthaxanthin, dicyano-β-carotene, zeaxanthin and lutein)	BM	Inclusion complex	Artificial light-harvesting, photoredox, and catalytic devices; Drug delivery	[20,25]
Hydroxypropyl β-cyclodextrin	Carotenoids (astaxanthin), glyceryl monostearate	BM	H-bonds	Drug delivery	[21]
Hydroxypropyl β-cyclodextrin	Simvastatin; atorvastatin; olmesartan medoxomil, N-methyl-D-glucamine	BM	Inclusion complex	Drug delivery	[29,30,34]
Arabinogalactan	Inositol hexanicotinate, glycyrrhizic acid	BM	Self-micelle solid dispersion	Drug delivery	[35]
Arabinogalactan	Ibuprofen; nifedipine; nimesulide; warfarin; valsartan	BM	Inclusion complex	Drug delivery	[22,23,28,36,38]
Arabinogalactan	Salicylic and acetylsalicylic acid; curcumin	BM	Inclusion complex	Drug delivery	[24,27,31,33]
Arabinogalactan	Kaempferol; betulin dipropionate; rimantadine; cyclophosphamide	BM	H-bonds	Drug delivery	[32,42,43,44,45]
Arabinogalactan	Pesticides: tebuconazole, imidacloprid, imazalil, prochloraz, chlorsulfuron, carbendasim, and thiram	BM	H-bonds	Plant protection	[47,48,49,50]
Lignocellulosic biomass	Niobium oxide, zinc oxide	BM	Nanocomposites	Nanocatalysts	[66,67]
Hyaluronic acid	Boron	TSE	Nanocomposites	Formulations for boron-neutron capture therapy	[100]
Chitosan	Trypsin	Bridgman anvil	H-bonds	Enzyme immobilization	[96]
Lignocellulosic biomass, sphagnum moss peat	Alkalis, enzymes or without additives	BM	Polysaccharide and humic fractions	Partially water-soluble products having antioxidant and catalytic activity	[57,58,59]
Birch wood, cellulose	Acids, alkalis, and enzymes	TSE	Cellulose, mercerized cellulose	Delignification, mercerization	[82,83]
Chitin	Alkali	TSE	Chitosan	By itself, as intermediate	[80,84,85]
Chitosan	Dicarboxylic acids and anhydrides; hydroxymethyl propionic acid	TSE	Acylated derivatives	Covalent and ionic hydrogels, amphiphilic derivatives	[86,87]
Cellulose, chitin (chitosan)	Sodium hydroxide; diglycidyl ether of oligo (ethylene oxide)	TSE	H-bonds, blends	Sorbents	[88]
Chitosan	Allyl bromide	TSE	Unsaturated derivatives	Photosensitive components in laser stereolithography	[89,90]
Chitosan	Docosylglycidyl and hexadecylglycidyl ethers	TSE	Alkylated hydrophobic derivatives	Rheological modifiers, selective sorbents and stabilizers	[91]
Chitin	Sodium hydroxide, polyvinyl acetate	TSE	Partially water-soluble chitosan-g- polyvinyl alcohol copolymers	Stabilizers of inorganic nanoparticles and biologically active substances	[80,93]
Chitosan/hydroxyethyl cellulose	Sodium hydroxide	TSE	Partially water-soluble chitosan-g- hydroxyethyl cellulose copolymers	Stabilizers of inorganic nanoparticles and biologically active substances	[92]
Chitosan	Maleic anhydride, polyethylene	TSE	Blends	Selective sorbents	[94]

^1^ Possibilities of mechanochemical modification of the polysaccharides in solid state using impulse (Ball milling (BM)) or continuous (Twin-screw extrusion (TSE), Bridgman anvil) impact.

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
