# Peer review of "Mechanochemical Transformations of Polysaccharides: A Systematic Review"

_ijms, 2022, doi:10.3390/ijms231810458_

Round 1

Reviewer 1 Report

This manuscript systematically reviewed the trends and results of studies on the modification of polysaccharides under mechanochemical conditions. This topic is interesting and suitable for publication on your journal. It is well-written and can be accepted after a minor modification.

The grammatical errors throughout the manuscript should be improved, such as the text in lines 122-123, 162, 175

As described in lines 175-176, the formation of hydrogen bonds is observed only after mechanochemical treatment for a long time. Pls add more details.

Author Response

This manuscript systematically reviewed the trends and results of studies on the modification of polysaccharides under mechanochemical conditions. This topic is interesting and suitable for publication on your journal. It is well-written and can be accepted after a minor modification.
The grammatical errors throughout the manuscript should be improved, such as the text in lines 122-123, 162, 175

Answer: Thank you for the comments! We revised the manuscript and corrected the grammatical errors.

As described in lines 175-176, the formation of hydrogen bonds is observed only after mechanochemical treatment for a long time. Pls add more details.

Answer: Thank you for the comments! This passage was revised.

Reviewer 2 Report

The review paper submitted for consideration in the journal IJMS by T.A. Akopova, T.N. Popyrina, and T.S. Demina is relatively correct, but it is a pity that it was narrowed down to authors from only one country. However, I understand that the manuscript is intended for a special issue that takes into account just the contributions of researchers from one country. It is therefore justified. 

I have the following comments on the manuscript for consideration. I believe that the "Results and discussion" chapter is intended for research papers. For review articles, such a chapter is unjustified.  It would be useful to divide the main part of the article, precisely that which the authors called "Results and discussion" into subsections. This would make it easier to read and understand the authors' intentions. As it is I see only one main idea of the paper, the contribution of Russian researchers. In the Introduction (l. 24 and 25), "an increase in the availability of functional groups" is a jargon phrase and should be corrected as such. Also in the Introduction (l. 44-48), this passage of text, as directing to one experimental work, is unjustified in this part of the manuscript. The flow of thought in the Introduction is disturbed by placing it here. In "Methods" (l. 72-79) should not include information about the role of individual authors in the creation of the paper. The publisher recommends placing this type of information at the end of the manuscript. It is better to insert the reference numbers immediately after the authors' names (l. 159 and 172), rather than after the description of the individual studies. 

Author Response

The review paper submitted for consideration in the journal IJMS by T.A. Akopova, T.N. Popyrina, and T.S. Demina is relatively correct, but it is a pity that it was narrowed down to authors from only one country. However, I understand that the manuscript is intended for a special issue that takes into account just the contributions of researchers from one country. It is therefore justified. 

Answer: Thank you for the comment. Indeed, this review belongs to the topical collection "State-of-the-Art Macromolecules in Russia", so we have to narrow it down to authors affiliated to universities/institutes of only one country. However, the found trends are in a mainstream of mechanochemistry research worldwide.

I have the following comments on the manuscript for consideration. I believe that the "Results and discussion" chapter is intended for research papers. For review articles, such a chapter is unjustified. It would be useful to divide the main part of the article, precisely that which the authors called "Results and discussion" into subsections. This would make it easier to read and understand the authors' intentions. As it is I see only one main idea of the paper, the contribution of Russian researchers.

Answer: We have to keep "Results and discussion" chapter in order to meet Systematic Review format per instructions of the Editorial office. However, we completely agree with the Referee and added subsections to make it easier to follow the manuscript rationale and catch the main idea of the paper.

In the Introduction (l. 24 and 25), "an increase in the availability of functional groups" is a jargon phrase and should be corrected as such.

Answer: The phrase was corrected.

Also in the Introduction (l. 44-48), this passage of text, as directing to one experimental work, is unjustified in this part of the manuscript. The flow of thought in the Introduction is disturbed by placing it here.

Answer: We agree with the Referee. This passage was removed.

In "Methods" (l. 72-79) should not include information about the role of individual authors in the creation of the paper. The publisher recommends placing this type of information at the end of the manuscript.

Answer: Thank you for the comment! We removed the names of authors from this section. The role of individual authors is described at the end of the manuscript per journal instructions.

It is better to insert the reference numbers immediately after the authors' names (l. 159 and 172), rather than after the description of the individual studies. 

Answer: The reference numbers are inserted immediately after the authors' names in the revised version of the manuscript.

Reviewer 3 Report

The manuscript presents an interesting approach related to the topic of polysaccharides. At some points, the text becomes difficult to understand, the inclusion of figures, tables, or diagrams would help to understand the information. Being a review article, it is important that the authors include tables that allow the results of the different scientific studies to be compared. Some suggestions are: to include an image of mechanical activation of natural polysaccharides; to include images of modification of polysaccharides chemical structure and fabrication of composite materials.

Author Response

The manuscript presents an interesting approach related to the topic of polysaccharides. At some points, the text becomes difficult to understand, the inclusion of figures, tables, or diagrams would help to understand the information. Being a review article, it is important that the authors include tables that allow the results of the different scientific studies to be compared. Some suggestions are: to include an image of mechanical activation of natural polysaccharides; to include images of modification of polysaccharides chemical structure and fabrication of composite materials.

Answer: Thank you for suggestions! We have added illustrative material (Figure and Table) to facilitate understanding of the material.

Round 2

Reviewer 3 Report

The authors have addressed my concerns and improved the manuscript.